# Updated Pharmacological Effects, Molecular Mechanisms, and Therapeutic Potential of Natural Product Geniposide

**DOI:** 10.3390/molecules27103319

**Published:** 2022-05-21

**Authors:** Liping Liu, Qin Wu, Yuping Chen, Guoxiang Gu, Runan Gao, Bo Peng, Yue Wang, Anbang Li, Jipeng Guo, Xinru Xu, Xiaochen Shao, Lingxing Li, Ya Shen, Jihu Sun

**Affiliations:** 1School of Pharmacy, Jiangsu Vocational College of Medicine, #283 Jiefang South Road, Yancheng 224000, China; liuliping1487@163.com (L.L.); g0105012022@163.com (R.G.); pbxl18352221037@163.com (B.P.); mxuanyuer@163.com (Y.W.); 18360718363@163.com (A.L.); 18351461883@163.com (J.G.); x18402583040@163.com (X.X.); chelsea020928@163.com (X.S.); llx1808470244@163.com (L.L.); 19516563139@139.com (Y.S.); 2Medical School, Jiangsu Vocational College of Medicine, #283 Jiefang South Road, Yancheng 224000, China; hhwuq@163.com (Q.W.); ggxtcm86@163.com (G.G.); 3Department of Basic Medical Science, Jiangsu Vocational College of Medicine, Yancheng 224005, China; chenyuping1985@126.com; 4Institute of Biotechnology, Jiangsu Vocational College of Medicine, #283 Jiefang South Road, Yancheng 224000, China

**Keywords:** geniposide, pharmacological effects, pharmacokinetics, toxicity

## Abstract

At present, the potential of natural products in new drug development has attracted more and more scientists’ attention, and natural products have become an important source for the treatment of various diseases or important lead compounds. Geniposide, as a novel iridoid glycoside compound, is an active natural product isolated from the herb *Gardenia jasminoides* Ellis (GJ) for the first time; it is also the main active component of GJ. Recent studies have found that geniposide has multiple pharmacological effects and biological activities, including hepatoprotective activity, an anti-osteoporosis effect, an antitumor effect, an anti-diabetic effect, ananti-myocardial dysfunction effect, a neuroprotective effect, and other protective effects. In this study, the latest research progress of the natural product geniposide is systematically described, and the pharmacological effects, pharmacokinetics, and toxicity of geniposide are also summarized and discussed comprehensively. We also emphasize the major pathways modulated by geniposide, offering new insights into the pharmacological effects of geniposide as a promising drug candidate for multiple disorders.

## 1. Introduction

Traditional Chinese medicines (TCMs) have a long history of applications in disease control in China. At present, with the development of chemical and pharmacological technologies, the application of natural products separated from TCMs is gradually being realized through orderly and standardized systems, bringing new insights into drug discovery [1,2,3]. Numerous studies have shown that natural products have significant therapeutic effects on various diseases. For instance, arsenic trioxide extracted from mineral arsenic could be used as an immunomodulator in the treatment of acute myeloid leukemia [4], and artemisinin derived from the plant *Artemisia annua* could be used to treat malaria [5]. Moreover, baicalin extracted from *Scutellaria baicalensis* Georgi could be used to treat liver and gut diseases [6]. In general, natural products may be an important source for the treatment of various diseases and new drug development.

Iridoid glycosides are active natural products isolated from TCMs and have extensive pharmacological and physiological effects [7]. Iridoid glycosides extracted from various TCMs have significant curative effects in the treatment of diabetes, cardiovascular diseases, neurological diseases, and cancer [8,9,10]. As a widely used TCM, *Gardenia jasminoides* Ellis (GJ) has been used for treating diabetes, jaundice hepatitis, hypertension, sprains, and contusions [11]. Geniposide is an iridoid glycoside compound that has been isolated from GJ fruit for the first time [12]. At present, geniposide has been found in more than 40 kinds of plants, most of which come from the *Rubiaceae* family, such as GJ, *Eucommia ulmoides* Oliver, *Rehmannia glutinosa* (Gaetn.) Libosch. ex Fisch. et Mey, and other TCMs.

The content of geniposide from different sources is very different. GJ is the most important source of geniposide, containing a very high content of geniposide (about 3.3–8.56%); geniposide is also the major phytochemical marker for the quality control of GJ [13]. According to relevant literature, about 90 kinds of geniposide derivatives have been found in natural plants, some of which are substituent derivatives of different parts of geniposide, such as daphylloside and geniposidic acid, and others that are structural isomer derivatives, such as monotropein, majoroside, and alpinoside. In addition, there are some biosynthetic products of geniposide, such as gardenoside and 6α/β-hydroxygeniposide [13]. 

In order to obtain geniposide as a more natural product, we systematically researched the main synthetic methods of producing geniposide and found that phosphine catalyst was helpful in generating geniposide. In brief, ethyl 2,3-butadienoate and enone (*S*)-3b underwent a [3+2] cycloaddition reaction under phosphine catalysis to generate cis-fused cyclopenta[c]pyran 4; then, in 10 steps, the cis-fused cyclopenta[c]pyran 4 was converted to the natural product iridoid glycoside (+)-geniposide (Figure 1) [14]. Previous literature reviews on geniposide highlighted its antitumor effect [15], antidiabetic effect [16], neuroprotective effect [17], and hepatoprotective and cholagogic effects [13]. Most of these pharmacological activities of geniposide are associated with its anti-inflammatory and antioxidant efficacies. With the advances of pharmacological research, geniposide has been considered as a promising multitarget drug (MTD) for treating multiple diseases. 

In this review, we present the recent discoveries of natural product geniposide associated with its therapeutic potential, its pharmacokinetics, and its toxicity. This review also highlights the potential molecular mechanisms of geniposide in the prevention and treatment of different diseases, and provides our comments about the omnipotent effects of geniposide.

## 2. Hepatoprotective Effect of Geniposide

Chronic liver disease (CLD) is a common chronic disease accompanied by persistent inflammation and fibrogenesis [18]. Liver fibrosis is a complex wound-healing response and a key driver of CLD, because it is a common pathological feature of liver failure [19]. A study showed that, in vivo, geniposide could significantly decrease liver enzymes levels in mice with liver fibrosis, improve the pathological morphology of mice with liver fibrosis, increase the activities of superoxide dismutase (SOD) and glutathione peroxidase (GSH-Px), and reduce the malondialdehyde (MDA) level in the liver. Geniposide can also inhibit oxidative stress, reduce liver inflammation and hepatocyte apoptosis, and regulate the metabolism of glycerophospholipid, arginine, proline, and arachidonic acid to treat liver fibrosis. These findings suggested that geniposide exerted an anti-hepatic fibrosis effect by inhibiting oxidative stress [20]. 

Another study showed that geniposide might also significantly improve liver fibrosis injury by inhibiting TGF-β1/Smad signaling in a CCl-induced BALB/c mouse model [21]. Geniposide could also ameliorate ccl4-induced liver fibrosis injury by modulating metabolic pathways, such as glycolysis, glutathione, and sulfur metabolism [22]. In vitro, geniposide significantly inhibited the sonic hedgehog (SHH) signaling pathway and reduced cell viability via downregulating the level of protein α-smooth muscle actin (α-SMA), causing G2/M cell arrest after HSC-T6 cell activation, thereby inhibiting the proliferation and activation of HSC-T6 cells via the SHH signaling pathway. These results suggested that geniposide exerted an anti-hepatic fibrosis effect through inhibiting the SHH signaling pathway [23]. 

Alcoholic liver disease (ALD) is the primary cause of CLD in western countries [24]. In the United States, more than 2 million people suffer from alcohol-related cirrhosis; ALD is also a key cause of liver injury-induced death and indications for liver transplantation [25]. In vivo, geniposide ameliorated acute alcohol-induced liver damage in mice by promoting the expression of the main antioxidant enzymes. These results indicated that geniposide significantly reversed the excessive elevation of serum hepatic lipid peroxidation (LPO) and the level of alanine/aspartate transaminase (ALT/AST) induced by alcohol, thus improving alcohol-induced liver oxidative stress damage [26]. Geniposide alleviated alcoholic steatosis and liver damage in rats via improving the transcriptional activity of peroxisome proliferator-activated receptor-α and hepatocyte nuclear factors 1α and 4α [27]. 

Serum metabolomic analysis of ALD model BALB/c mice showed that geniposide could regulate 21 metabolic pathways related to ALD, including amino acid metabolism, pyruvate metabolism, and the TCA cycle. Amino acid metabolism is the most significant of these pathways, suggesting that geniposide may improve ALD mice by targeting amino acid metabolism [28]. In vitro, geniposide reduced the miR-144-5p level, directly acting on isocitrate dehydrogenase (IDH)1and IDH2, and improved the function of Enmh cells by modulating the TCA cycle in an ethanol-induced hepatocyte apoptosis model, thereby ameliorating alcohol-induced liver injury [29]. Furthermore, geniposide alleviated alpha-naphthylisothiocyanate (ANIT)-induced hepatotoxicity in mice by inhibiting the activation and expression of NF and STAT3 [30]. In Figure 2, we summarize the mechanism of geniposide in targeting liver injury diseases.

## 3. Effect of Geniposide on Osteoporosis

Osteoporosis (OP) is a systemic bone metabolic disease associated with bone mineral loss and reduced strength, causing an increased risk of bone fractures [31]. In vivo, geniposide significantly improved the pathological damage of trabecular bone in DEX-induced OA model rats and improved the OA model-induced increase in ATF4/CHOP expression. In addition, geniposide inhibited DEX-induced mitochondrial apoptosis in MC3T3-E1 cells, an inhibitory effect that may be related to geniposide’s activation of NRF2 expression and reduction of ER stress [32]. In vitro, geniposide attenuated the inhibitory effect of dexamethasone (DEX)-induced osteogenic differentiation by activating the expression of the glucagon-like receptor (GLP-1R). Geniposide promoted alkaline phosphatase (ALP) activity and mineralization, the mRNA and protein expression of osteopontin (OPN), Runt-related transcription factor 2 (Runx2), and Osterix (Osx), and activated the extracellular regulated protein kinase (ERK) pathway in MC3T3-E1 cells treated with DEX. These results indicated that geniposide ameliorated glucocorticoid-induced osteogenesis by inhibiting MC3T3-E1 cells [33]. Geniposide promoted the proliferation and differentiation of osteoblasts MC3T3-E1 and ATDC5 cells by targeting microRNA-214 to activate the Wnt/β-catenin signaling pathway [34]. In addition, geniposide ameliorated DEX-induced cholesterol accumulation in MC3T3-E1 cells and inhibited cell differentiation through modulating the GLP-1R/ABCA1 axis. Moreover, geniposide improved DEX-induced OP by activating GLP-1R expression in vivo and in vitro [35]. In addition, geniposide has the function of anti-skeletal muscle fibrosis. Geniposide significantly reduced the expression of pro-fibrotic genes caused by TGF-β and Smad4 in vitro. Furthermore, geniposide significantly downregulated the pro-fibrotic genes to improve skeletal muscle injury in a mouse contusion model in vivo [36]. In Figure 3, we summarize the mechanism of geniposide in targeting osteoporosis.

## 4. Antitumor Effect of Geniposide

Traditional Chinese medicine therapy can be used as adjuvant therapy for cancer treatment without obvious side effects on the human body. Many studies have reported that geniposide exerts broad anticancer effects. In vivo, geniposide significantly disrupted the proliferation and invasion of hepatocellular carcinoma (HCC), angiogenesis, and lung metastasis in orthotopic HCC mice. The direct inhibition of geniposide on TLR4/MyD88 activation contributes to the inhibition of STAT3/SP1-dependent vascular endothelial growth factor (VEGF) overexpression in HCC angiogenesis and lung metastasis. However, this contribution is not affected by HIF-1αstability [37]. Geniposide can improve aflatoxin B (1)-induced early hepatocarcinogenesis in rats by inhibiting glutamyltranspeptidase activity [38]. 

In vitro, geniposide can inhibit the proliferation and induce apoptosis of human breast cancer MCF-7 cell in 2D and 3D cultures [39]. Geniposide significantly inhibited oral squamous carcinoma HSC-3 cells by upregulating the expression of caspase-3, caspase-8, caspase-9, Bax, p53, P21, IκB-α, Fas, FasL, TIMP-1, and TIMP-2, and downregulating the expression of Bcl-2, Bcl-xL, HIAP-1, HIAP-2, NF-κB, COX-2, iNOS, MMP-2, and MMP-9 [40]. Geniposide also promoted cell apoptosis by inhibiting the proliferation, migration, and invasion of HepG2 and Huh7 liver cancer cells, and this antitumor effect was mainly achieved by targeting miR-224 to block the Wnt/β-catenin and AKT cascades [41]. Furthermore, geniposide inhibited the viability, colony formation, migration, and invasion of MNK45 cells by downregulating HULC. Geniposide inactivated PI3K/AKT and JNK signaling pathways by downregulating HULC [42]. Geniposide also significantly inhibited diffuse large B-cell lymphoma (DLBCL) cell proliferation and promoted apoptosis; this effect was at least partially mediated through the HCP5/miR-27b-3p/MET axis [43]. Moreover, geniposide significantly inhibited the proliferation of medulloblastoma cells and promoted apoptosis; thus, this anticancer effect was mainly achieved by downregulating the expression of miR-373 to block the Ras/Raf/MEK/ERK pathway [44]. In Figure 4, we summarize the mechanism of geniposide in targeting tumors.

## 5. Anti-Diabetic Effect of Geniposide

In recent years, the prevalence of diabetes has continuously grown. The World Health Organization has reported that diabetes increases the risk of premature death and possible diabetes-related complications, such as cardiovascular risk, cognitive dysfunction, and kidney failure [45]. In order to improve patients’ compliance, long-acting hypoglycemic drugs need to be developed. In vivo, geniposide significantly reduced the levels of blood glucose, insulin, and TG in diabetic mice, and reduced the expression of hepatic glycogen phosphorylase (GP) and glucose-6-phosphatase (G6Pase) at mRNA and immunoreactive protein levels, as well as enzymes’ activity. These results indicated that geniposide may reduce blood glucose by inhibiting the expression of GP and G6Pase [46]. Geniposide ameliorated abnormal lipid metabolism mice by inhibiting body weight, visceral fat accumulation, and intrahepatic lipid accumulation in spontaneously obese type 2 diabetic TSOD. Furthermore, geniposide helped with abnormal glucose tolerance and hyperinsulinemia to alleviate insulin resistance [47]. Geniposide interfered with the synthesis, secretion, and homeostasis of RBP4 in the circulation of high-fat diet mice, promoting skeletal muscle glucose uptake and reducing glycogen storage, thereby improving systemic insulin sensitivity [48]. A study showed that geniposide also played a crucial role in accelerating β-cell survival and regeneration. Geniposide accelerated β-cell regeneration and blood glucose normalization in high-fat diet and db/db mice. At the same time, geniposide improved pancreatic β-cell proliferation in diabetic mice, induced ductal cell differentiation by accelerating the expression of TCF7L2, and activated the JAK2/STAT3 signaling pathway in mouse pancreatic exocrine cells. Moreover, geniposide improved β-cell survival by accelerating proliferation and reducing apoptosis in cultured mouse islets following diabetic stimulation [49]. In addition, geniposide promoted wound healing in diabetic rats by anti-inflammatory and blood glucose regulation [50].

In vitro, geniposide improved glucose homeostasis by inhibiting the FoxO1/PDK4 signaling pathway [51]. Geniposide promoted insulin secretion in rat INS-1 insulinoma cells by activating the GLP-1 receptor [52]. By constructing the insulin resistance (IR) model of HepG2 cells, it was determined that geniposide may accelerate autophagy and inhibit IR of HepG2 cells through P62/NFκ-B/Glut-4 [53]. Geniposide enhanced insulin secretion by activating GLP-1R and the adenylate cyclase (AC)/cAMP signaling pathway [54]. Geniposide modulated acute stimulation-induced ATP production and glucose-stimulated insulin secretion (GSIS) in pancreatic islet beta cell lines at high glucose concentrations. Geniposide protected rat insulinoma cells from apoptosis at a high glucose concentration, mainly due to promoting the apoptosis-related Bcl-2/BAX protein ratio. Furthermore, geniposide dose-dependently improved the function and proliferation of β-cell under chronic hyperglycemia conditions [55]. 

Numerous studies have indicated that geniposide also has a significant curative effect on diabetic complications. In vivo, geniposide significantly reduced body weight, blood glucose, and cognitive decline in diabetic mice. Geniposide also inhibited the production of inflammatory cytokines. In addition, geniposide improved the changes in BTK, TLR4, MyD88, NF-κB, and the brain-derived neurotrophic factor (BDNF) induced by diabetic mice. These results suggested that geniposide may ameliorate STZ-induced cognitive impairment by regulating the BTK/TLR4/NF-κB signaling pathway [56]. Geniposide also improved diabetic depression. It was found that geniposide attenuated depression-like behaviors caused by diabetes via upregulating the mRNA expression of BDNF and tropomyosin-related kinase B (TrkB) in the hippocampus of diabetic mice [57]. Furthermore, geniposide significantly improved diabetic nephropathy by ameliorating the renal structural and functional abnormalities via inhibiting the NF-κB-mediated inflammatory response in diabetic rats [58]. In Figure 5, we summarize the mechanism of geniposide in targeting diabetes.

## 6. Anti-Myocardial Dysfunction of Geniposide

As an agonist of GLP-1R, geniposide could protect cardiac hypertrophy by activating the GLP-1 receptor/AMPKα pathway, thereby being a potential candidate drug for treating cardiac hypertrophy [59]. In vivo, geniposide activated AMPKα to inhibit the accumulation of myocardial reactive oxygen species (ROS) in a mouse model of LPS-induced sepsis-induced myocardial dysfunction, thereby blocking the apoptosis and pyroptosis of myocardial cells mediated by NOD-like receptor protein 3(NLRP3), and ameliorating sepsis-induced myocardial dysfunction [60]. Geniposide inhibited oxidative stress, endoplasmic reticulum stress, acetylated Smad3 (ac-Smad3), and the P-Samd3 pathway independent of SIRT1 activation, thereby antagonizing isoproterenol (ISO)-induced cardiac fibrosis in mice. These findings suggested that geniposide might be a potential anti-myocardial fibrosis drug [61]. In addition, geniposide attenuated the myocardial inflammatory response and cardiomyocyte apoptosis caused by HFD in mice. Geniposide also activated AMPK and sirtuin (Sirt1) in vivo and in vitro. These findings suggested that geniposide reversed HFD-induced cardiac dysfunction through an AMPK- and sirt1-dependent mechanism and that it had a protective effect against obesity-related cardiac injury. The above results showed that geniposide might be a potential drug for the treatment of obesity-induced cardiovascular complications [62]. In the NaCl solution-induced myocardial injury model of spontaneously hypertensive rats, geniposide significantly reduced COI and COIII levels, promoted p-AMPK, and heightened energy metabolism pathways, thus improving myocardial injury. In addition, this protective effect may have been achieved by modulating the AMPK/SirT1/FOXO1 energy metabolism signaling pathway [63]. In a rat model of myocardial ischemia-reperfusion (I/R) injury, geniposide significantly reduced the size of myocardial infarction, improved cardiac function, and inhibited cell apoptosis. Geniposide also inhibited the expression of autophagy-related proteins and the accumulation of autophagosomes in vitro and in vivo, thus improving myocardial injury [64]. Moreover, this cardioprotective effect may be related to the activation of the AKT/mTOR signaling pathway to inhibit autophagy. In addition, geniposide may have improved myocardial ischemia-reperfusion injury in diabetic rats by inhibiting oxidative stress via activating the NRF2/HO-1 signaling pathway [65].

In vitro, geniposide inhibited cardiomyocyte apoptosis caused by hypoxia/reoxygenation (H/R) by reversing mitochondrial dysfunction and might be correlated with the activation of the GLP-1R and PI3K/AKT signaling pathways [66]. In Figure 6, we summarize the mechanism of geniposide in targeting myocardial dysfunction.

## 7. Neuroprotective Effect of Geniposide

Geniposide is easily soluble in water and has significant anti-inflammatory activity, which could exert neuroprotective effects through the blood-brain barrier (BBB). In vivo, geniposide maintained the integrity of the blood-brain barrier and alleviated brain edema in traumatic brain-injury (TBI) rats. Geniposide significantly inhibited the pro-inflammatory factors levels of IL-1β, IL-6, and IL-8 in the plasma of TBI rats, upregulated the level of IL-10 anti-inflammatory factors, and inhibited the activities of p-p38 mitogen-activated protein kinase (p-p38) and p-NF-κB p65 (p-p65). These results indicated that geniposide exerted neuroprotective effects on TBI by inhibiting the activities of p-p38 and p-p65 [67]. In a mouse model of hypoxic-ischemia (HI)-induced brain injury, geniposide significantly inhibited apoptosis in brain tissue, inhibited the proliferation of immune cells (microglia and astrocytes), and reduced IgG leakage, thus improving brain injury; this protective effect may be closely related to the activation of the PI3K/Akt signaling pathway [68].

Another potential risk of secondary TBI injury is impairment of the integrity of the BBB, which can lead to leaky blood vessels, hypoxia, and edema. In vitro, a BBB model was jointly constructed by brain microvascular endothelial cells (BMECs) and astrocytes. Geniposide reduced BBB permeability, promoted the expression of tight junction proteins (occludin-1, claudin-5, and occludin) and γ-glutamyl transpeptidase, and increased trans-endothelial cell resistance, thereby effectively improving the BBB function. Geniposide also reduced oxidative stress injury and the release of pro-inflammatory factors, increased the release of BDNF and glial cell-derived neurotrophic factors, and downregulated the expression of matrix metallopeptidases-9 (MMP-9) and MMP-2. These results suggested that geniposide can improve the BBB dysfunction induced by oxygen glucose deprivation and reoxygenation (OGD/R) through multiple protective mechanisms [69]. It is well-known that GJ is often used to treat stroke and infectious diseases in TCM clinical practice, while geniposide is the critical active ingredient of GJ against ischemic stroke. An in vitro model of cerebral ischemia in BMECs was established by OGD. Geniposide suppressed the downstream ERK1/2 signaling pathway and the release of pro-inflammatory factors IL-8, IL-1β, and MCP-1 by interfering the expression of the P2Y14 receptor, thereby exerting anti-inflammatory effects, which provided a basis for the clinical application of geniposide in cerebral ischemia [70]. Geniposide can also protect neurons from post-ischemic vascular injury by activating the GluN2A/AKT/ERK pathway [71]. Geniposide inhibited the apoptosis of primary cultured hippocampal neurons caused by 3-morpholinosydnonimine hydrochloride (SIN-1) by upregulating the expression of heme oxygenase 1 (HO-1). Geniposide induced nuclear translocation of nuclear factor-E2-related factor 2 (Nrf2) and activated phospatidylinositol 3’-kinase (PI3K) under oxidative stress conditions, whereas zinc protoporphyrin (HO-1 inhibitor) and LY294002 (a specific inhibitor of PI3K) reduced the protective effect of geniposide on hippocampal neurons. These results illustrated that the novel cytoprotective mechanism of geniposide against oxidative stress might be associated with upregulating the expression of HO-1 mediated by PI3K and Nrf2 [72]. Geniposide also enhanced the expression of HO-1 via the cAMP-PKA-CREB signaling pathway, thereby protecting PC12 cells from SIN-1 induced-oxidative injury. In addition, GLP-1R exhibited a crucial regulatory role in the promotion of HO-1 expression by geniposide, thereby mitigating oxidative damage in PC12 cells [73].

Moreover, geniposide has a good protective effect on neurodegenerative diseases such as Parkinson’s disease (PD) and Alzheimer’s disease (AD). In vivo, geniposide restored the quantity of tyrosine hydroxylase-positive dopaminergic neurons in the substantia nigra pars compacta in a PD mouse model, increased the Bax level, reduced the apoptosis signaling molecule Bcl-2 level, and activated caspase 3 in substantia nigra. The above results indicated that geniposide exerted a neuroprotective effect by improving growth factor signal transduction and lowering apoptosis [74]. Geniposide exerted neuroprotective effects on inhibiting the expression of α-synuclein via the miR-21/LAMP2A axis in a PD mouse model [75]. Recent studies have shown that mitochondrial dysfunction promotes the development of AD. It has been found that geniposide defended against Aβ-mediated mitochondrial dysfunction of primary cortical neurons in an AD transgenic mouse model by recovering ATP production and the activity of mitochondrial membrane potential (MMP), cytochrome c oxidase (CcO), and caspase 3/9 activity, reducing ROS generation and cytochrome c leakage, and inhibiting apoptosis. These results suggested that geniposide attenuated Aβ-induced neuronal damage in an AD transgenic mouse model by inhibiting oxidative stress and mitochondrial dysfunction [76]. Furthermore, geniposide inhibited Aβ-rage interaction-mediated over-activation of MAPK signaling, thereby reducing Aβ accumulation and ameliorating cholinergic deficits in the hippocampus in middle-aged Alzheimer model mice. Geniposide also decreased AChE activity by increasing ChAT levels and activity in primary hippocampal neurons and reducing the toxic effects of Aβ 1–42 oligomer-induced cholinergic depletion. These results indicated that geniposide enhanced the cholinergic neurotransmitter, which may be one of the reasons for its memory-enhancing effect [77]. Geniposide also has beneficial effects on diabetic AD. In a streptozotocin-induced Alzheimer rat model, geniposide promoted the expression of GSK3β (PS-9) and inhibited the expression of GSK3β (PY-216), suggesting that geniposide inhibited the overactivity of GSK3β induced by streptozotocin (STZ) in an STZ-induced Alzheimer rat model. Furthermore, ultrastructural analysis showed that geniposide protected against streptozotocin (STZ)-induced neuropathology, including paired helical filament (PHF)-like structures, accumulation of vesicles at synaptic terminals, endoplasmic reticulum (ER) abnormalitiess, and stages of apoptosis. The above results indicated that geniposide might be a potential drug candidate for treating sporadic AD [78]. Geniposide significantly improved repeated restraint stress (RRS)-induced depression-like behavior in mice, inhibited hippocampal neuronal apoptosis, and reduced the proinflammatory cytokine IL-1β and TNF-α levels. Furthermore, geniposide restored the GLP-1R/protein kinase B (AKT) signaling-related protein. These results suggested that the antidepressant-like effect of geniposide might be strongly connected with the GLP-1R/AKT signal transduction [79]. Geniposide also upregulated Six3os1 by regulating miR-511-3p/Fezf1/AKT axis, thereby ameliorating oxidative stress in mice with chronic unpredictable mild stress (CUMS)-induced depression-like behavior [80]. In Figure 7, we summarize the mechanism of geniposide on neurological disease.

## 8. Other Therapeutic Effects of Geniposide

Age-related macular degeneration (AMD), predominantly wet AMD, is the leading cause of irreversible vision loss around the world, characterized by choroidal neovascularization (CNV). The stress or damage of retinal pigment epithelium (RPE) induces proangiogenic factors to drive CNV, while proangiogenic factors induced by the stress or damage of retinal pigment epithelium (RPE) contribute to drive CNV. Geniposide has been proven to have anti-angiogenic effects. Geniposide inhibited the transcription and expression of the heparin-binding epidermal growth factor in hypoxic RPE cells and a mouse laser-induced CNV model in vivo and in vitro. GLP-1R, as a geniposide receptor antagonist, inhibited the protective effect of geniposide. In addition, geniposide alleviated wet AMD by reducing the transcription and expression of the heparin-binding epidermal growth factor in hypoxia-exposed RPE cells by downregulating miR-145-5p/NF-κB axis [81]. In an ApoE-/- mouse model of high cholesterol diet-induced atherosclerosis, geniposide had immunomodulatory effects in preventing the formation of atherosclerotic lesions through declining the quantity of dendritic cells (DCs) and inhibiting the maturation and infiltration of DCs into lesions in the bone marrow [82]. Geniposide also improved atherosclerosis induced by ApoE-/- mice, upregulated the expression of foxp3, and accelerated the quantity and function of Treg cells by regulating lipids and the immune response to ameliorate the progression of atherosclerotic lesions [83]. Geniposide significantly restored spinal cord injury in rats, reduced spinal cord tissue edema, decreased the spinal cavity area, increased the number of nf-200 positive neurons and HRP-positive neurons, and regenerated axons with myelin sheath. Moreover, geniposide effectively alleviates inflammatory responses after spinal cord injury in rats by inhibiting the IKKs/NF-KB pathway, promoting motor function recovery and axon regeneration [84]. Geniposide also has a significant antithrombotic effect. Geniposide significantly inhibited thrombin/collagen-induced platelet aggregation and the venous thrombosis caused by tight ligation of the inferior vena cava in rats [85]. 

Geniposide also has a significant therapeutic effect on rheumatoid arthritis (RA) and osteoarthritis (OA). In vivo, geniposide attenuated the hyperpermeability of fibroblast-like synoviocytes (FLSs) in adjuvant-induced arthritis (AA) rats, possibly preventing RA by downregulating RhoA/p38MAPK/NF-κB/F-actin signaling. The above results suggested that geniposide might be a potential drug candidate for treating RA [86]. In an AA rat model, geniposide also significantly inhibited the pathological state of inflammation, reduced the secretion of VEGF and angiopoietin-1, promoted the secretion of endostatin, and inhibited FLSs hyperproliferative. Geniposide also significantly inhibited SphK1 activity, the S1P level, and the expression of SphK1 and S1PR1 in FLSs, indicating that geniposide reduced SphK1 activity by recovering the homeostasis of pro- and anti-angiogenic factors, thus disturbing SphK1-s1p-s1pr1 signaling transduction, eliminating synovial microangiogenesis and displaying anti-RA angiogenesis effect [87]. 

In vitro, geniposide alleviated RA by upregulating the expression of miRNA-124a, inhibiting cell proliferation treated with TNF-α, and blocking the activation of the Ras-Erk1/2 pathway [88]. Geniposide inhibited the expression of iNOS, COX-2, and MMP-13 in rat chondrocytes stimulated by IL-1β in vitro, while promoting the expression of type II collagen. Geniposide also increased the expression of anti-apoptotic protein Bcl-2 and inhibited the expression of pro-apoptotic proteins such as Bax, Cyto-c, and C-caspase3. These results indicated that the above-mentioned changes caused by geniposide were closely connected with inhibiting the activation of the PI3K/Akt/NF-κB pathway induced by IL-1β. Geniposide also inhibited the development of a rat OA model in vivo, suggesting that geniposide may be a promising drug candidate for treating OA [89]. Geniposide also inhibited the activation of PI3K-Akt signaling and upregulated the expression of phosphate and tension homology deleted on chromosome ten (PTEN), thus inhibiting RA angiogenesis and improving RA in vivo and in vitro [90]. 

In addition, geniposide has a significant anti-colitis effect. In vivo, geniposide promoted the phosphorylation of AMPK by downregulating the expression of NF-κB, COX-2, iNOS, and MLCK proteins, and upregulating the expression of tight junction proteins, such as occludin and ZO-1. These results suggested that geniposide improved the barrier dysfunction and TNBS-induced colitis in rats by inhibiting the AMPK-mediated MLCK pathway [91]. In Figure 8, we summarize other therapeutic effects of geniposide.

## 9. Pharmacokinetic Study of Geniposide

More and more studies have shown that there are significant differences in the pharmacokinetics of geniposide in different administration modes and different disease models. Geniposide was rapidly absorbed after peroral administration, with a peak plasma concentration after 1 h, and was rapidly eliminated from the plasma within 12 h. In addition, the absolute oral bioavailability of geniposide was low, at about 9.67%. Meanwhile, after the peroral administration of geniposide, the results of tissue distribution studies in rats showed that the concentration of geniposide was the highest in the kidney and the lowest in the brain [92]. In order to improve the problem of poor brain targeting by geniposide, the preparation of geniposide into geniposide liposome could significantly prolong the half-life of geniposide, enhance brain targeting, and better improve cerebral ischemia reperfusion injury (CIRI) rat [93]. The analysis of the pharmacokinetics of geniposide in adjuvant-induced arthritis (AA) rats by intragastric administration showed that the Cmax value and the average AUC of geniposide in the plasma of the high-dose group (120 mg/kg group) were significantly higher than those of the middle-dose group (60 mg/kg group) and the low-dose group (30 mg/kg group); the dose-dependent PK parameter t_1/2_ of each group was also significantly different. Interestingly, the inter-individual variability of the above three PK parameters was less than 40% in each instance, and the tmax of geniposide tended to be prolonged in the high-dose group. These results indicated that the absorption of geniposide is reliable [94]. After oral administration of different concentrations of geniposide, the concentration of geniposide was detected in the articular cavity microdialysate of AA rats, and it was found that the concentration of geniposide reached a maximum after 2 h of oral administration. Moreover, when the oral administered dose was 60 mg/kg, compared with other dose groups, the half-life was the longest and the elimination rate was the slowest [95]. Oral administration of geniposide in an adjuvant arthritis (AA) rat group and a normal rat group had significant differences in pharmacokinetic parameters. Compared with the normal rat group, the plasma geniposide concentration in the AA rat group was higher at most detection time points (especially 60–360 min), indicating that the elimination rate of geniposide in AA rats was slower [96]. A recent study found that in type 2 diabetic rats, the AUC of GJ fruit extract and pure geniposide increased similarly after oral administration, suggesting that the increase in AUC after gavage of GJ fruit extract was mainly due to the effect of geniposide, not to the other ingredients in GJ. These results suggested that the pathological state of diabetes altered the pharmacokinetics of geniposide, which may be a key factor contributing to the elevation of the AUC of geniposide [97]. In MCAO model rats, the pharmacokinetic results of geniposide alone or combined with baicalin or berberine showed that the combination with baicalin could improve the bioavailability of geniposide, while the combination with berberine decreased the bioavailability of geniposide. This effect may be related to the effect of baicalin inhibiting the hydrolysis of geniposide and berberine increasing the hydrolysis of geniposide. Meanwhile, there was a certain difference between normal rats and MCAO model rats in the pharmacokinetic study results of geniposide alone or combined with baicalin or berberine; the absorption effect of geniposide on MCAO rats (including shorter Tmax, higher apparent volume of distribution and Cmax, longer MRT) was better than that of normal rats [98]. 

The pharmacokinetic analysis of SD rats intragastrically administered with different doses of Yin-Chen-Hao Tang (YCHT) suspension showed that the Cmax value of the low-dose group (1 g/kg YCHT) was 0.145 ± 0.251 µg/mL, while the Cmax value for the high-dose group (3 g/kg YCHT) was 0.604 ± 0.256 µg/mL. Moreover, the ACU value of the high-dose group reached 4.88 ± 32.28 min μg/mL, which was 5.1 times that of the low-dose group. When adjusting the AUC to the administered dose, the ratio in the high-dose group was still 1.7 times that of the low-dose group [99]. Moreover, oral administration of YCHT in the acute liver injury model rat group and the normal rat group showed significant differences in the pharmacokinetic parameters of geniposide. Compared with the normal rat group, the Cmax and AUC0-∞ value of geniposide in the model group was higher, indicating that the geniposide significantly accumulated in the body under the disease state of liver injury [100]. The pharmacokinetics of geniposide in a compound Huanglianzhizi decoction or a Huanglianjiedu decoction in middle cerebral artery occlusion (MCAO) model rat analysis showed that compared with the control group, the AUC and Cmax of the two compound treatment groups showed a significant increase trend. As a result, the MRT data of MCAO was larger, indicating that under the pathological state of MCAO, the absorption of geniposide in the compound increased, the elimination rate slowed down, and the residence time in the body was longer [101]. In addition, the pharmacokinetic analysis of a Zhi-Zi-Hou-Pu decoction (ZZPHD) and GJ extract with equal doses of geniposide showed that ZZHPD significantly shortened the t_1/2_ of geniposide and increased the AUC of geniposide compared with GJ extract alone, suggesting that the GJ compound could significantly improve the oral bioavailability of geniposide compared with the use of GJ alone [102].

Finally, we further studied the metabolism of geniposide. The urine analysis results of geniposide metabolites in rats by intragastric administration showed that the main metabolic pathway of geniposide in vivo occurred after deglycosylation, followed by glucuronidation and the cleavage of the pyran ring. In vitro experiments further confirmed that the main metabolite of geniposide is the glucuronic acid conjugate of genipin [103]. β-glucosidase has good affinity and catalytic efficiency for geniposide and can hydrolyze geniposide to genipin [104], while genipin can be further metabolized to Genipin-1-*O*-glucuronic acid and genipin-monosulfate under the action of enzymes [105]. Interestingly, the pharmacokinetic parameters of these two metabolites were significantly different between the normal rat group and the diabetic model rat group. Compared with the normal rat group, the diabetic rat group had a longer Tmax and a significantly lower clearance rate, suggesting that the diabetes group had a longer mean residence time (MRT) and greater Cmax and AUC [106]. In addition, another study showed that a total of 33 metabolites of geniposide were found in SD male rats after oral administration of geniposide. The tissue distribution of geniposide metabolites showed that they were mainly distributed in rat heart (6 metabolites), liver (12 metabolites), spleen (3 metabolites), brain (6 metabolites), lung (6 metabolites), kidney (12 metabolites), and rat liver microsomes (RLMs) (4 metabolites). Moreover, the in vivo metabolism of geniposide mainly involved the hydrolytic pathway of C-1 and the pathway of a parent drug or isomer, including demethylation and glucosylation [107]. In AA rats, the tissue distribution of geniposide and its four main metabolites (G1-G4) was analyzed after oral administration of geniposide. The analysis results showed the plasma of AA rats mainly distributed geniposide, genipin (G1), the mono-glucuronide conjugate of genipin (G2); mesenteric lymph node (MLN) mainly distributed geniposide and G2; and spleen mainly distributed geniposide, G2, a cysteine conjugate ring-opened genipin (G3) and an oxidation of G3 (G4). GE, G1, G2, and G4 are mainly distributed in urine, and only geniposide was distributed in the liver and synovium [108].

## 10. Toxicity Study of Geniposide

Although a large number of studies have shown that geniposide has good pharmacological effects and biological activities, some recent studies have reported that high-dosage or continuous administration of geniposide may cause liver and kidney toxicity. Therefore, we studied the toxic effects of geniposide to provide theoretical guidance for the safe use of geniposide. The experimental results of continuous intragastric administration of rats with high-dose GJ water extract (30 g/kg) found significant hepatotoxicity. In addition, the higher the dose, the more serious the liver damage [109]. Another study confirmed that rats were given a high dose of GJ water extract (3.08 g/kg) or geniposide (280 mg/kg) by continuous intragastric administration for 3 days to cause significant liver damage. Further, the results of liver pathology showed obvious pathological features, such as hepatocyte swelling, necrosis, and inflammatory infiltration [110]. The results of metabolomic analysis of SD rats administered ZZHPD by gavage showed that for the high dose group of ZZHPD (27 g/kg/day) geniposide was significantly enriched in the liver, while the highest concentration was in the kidney, suggesting that geniposide may be the key cause of ZZHPD liver toxicity, and the kidney may be the main part of the excretion of geniposide [111]. Both geniposide and its aglycone genipin can significantly induce hepatotoxicity in rats. Moreover, the hepatotoxicity induced by the oral administration of geniposide (320 mg/kg) was comparable to that of intraperitoneal injection of genipin (80 mg/kg). Interestingly, the toxicity of geniposide was significantly enhanced when pretreated with sulfosulfoxide, whereas the toxicity of geniposide was completely suppressed by pretreatment with cysteine. The results indicated that the hepatotoxicity of geniposide was closely associated with the transformation of geniposide to genipin; the sulfhydryl in the liver was especially crucial in regulating the hepatotoxicity of geniposide [112]. The experimental results of continuous intragastric administration of rats with high-dose geniposide (300 mg/kg) showed significant nephrotoxicity, and the renal tubules showed mild pathological damage with mild swelling and vacuolar degeneration. Moreover, kidney injury molecule-1(Kim-1) and neutrophil hap-associated lipocalin(NGAL) were significantly increased in urine samples, indicating that significant nephrotoxicity would occur after continuous administration of large doses of gardenoside for 3 days [113]. In addition, another study showed that male and female SD rats that were given high-dose geniposide (100 mg/kg) by gavage for 26 weeks showed significant liver and kidney damage [114].

Although the above studies reported that high-dose and continuous administration of geniposide can induce liver and kidney injury, some studies have reported that low dose administration of GJ extract or geniposide has no toxic effect. A study showed that male and female SD rats were administered orally for 13 weeks with the yellow powder of GJ (containing 2.783% geniposide). The administered dose was equivalent to 60 mg/kg of geniposide without causing any serious toxic effects [115]. Another study showed that continuous oral administration of geniposide (24.3 and 72.9 mg/kg, respectively) for 90 days in male and female SD rats did not cause liver and kidney toxicity [116].

## 11. Conclusions

This work highlighted the current understanding of the pharmacological effects and associated mechanisms of geniposide. Geniposide is a natural active product with multiple potential health benefits, such as a hepatoprotective effect, an anti-osteoporosis effect, an antitumor effect, an anti-diabetes effect, an anti-myocardial dysfunction effect, a neuroprotective effect, and other beneficial effects. The mechanisms of action of geniposide mainly include anti-oxidation, anti-inflammation, and the regulation of apoptosis. These pharmacological activities often play an important part in the treatment of different diseases. For instance, geniposide has anti-inflammatory and anti-oxidative effects on liver injury, myocardial dysfunction, and nerve damage. Meanwhile, geniposide inhibits the hepatocyte apoptosis to ameliorate liver injury, and geniposide blocks the apoptosis of myocardial cells mediated by NLRP3 to ameliorate sepsis-induced myocardial dysfunction. The multiple effects of geniposide play a critical part in the treatment of diseases through the GLP-1R-mediated signaling pathway, the AMPK-mediated signaling pathway, the NF-κB signaling pathway, and other signaling pathways. These results show that geniposide has extremely high medicinal value.

Although numerous studies have elucidated the mechanisms by which geniposide has a wide range of pharmacological effects, some controversies still remain. For example, geniposide has protective effects on liver and diabetic nephropathy, while recent studies have reported that high doses or long-term administration of geniposide can cause mild swelling of renal tubules accompanied by vacuolar degeneration, and obvious swelling and necrosis of liver cells accompanied by inflammatory infiltration. These findings are controversial with the hepato-renal protective effect of geniposide [110,113,114]. Nonetheless, low-dose administration of GJ extract or geniposide had no toxic effects. A 3-month intake of GJ yellow powder containing 2.783% geniposide (nearly 60 mg/kg/day of geniposide) did not trigger any serious toxic response [115]. Moreover, the subchronic toxicity study indicated that geniposide did not give rise to hepatotoxicity after oral administration of 24.3 and 72.9 mg/kg for 90 days [116]. The results of the toxicity study of geniposide showed that proper dose of geniposide in the application process could ensure the safety and efficacy of clinical application. In addition, the pharmacokinetics of geniposide showed significant differences in different administration modes and different disease models. Therefore, in the process of treating diseases, we can rationally apply geniposide according to the metabolism and distribution of geniposide in each disease model.

In conclusion, geniposide achieves anti-hepatic injury effects, anti-osteoporosis effects, and other pharmacological effects by modulating multiple intracellular signaling pathways and targeting genes systematically. These effects fully reflect the systemic, multi-angle, and multi-target advantages of natural product geniposide in the treatment of various diseases. This review offers novel ideas for the design of effective drug targets at the molecular level and lays a theoretical foundation for further expanding the drug development and clinical applications of geniposide.

## Figures and Tables

**Figure 1 molecules-27-03319-f001:**
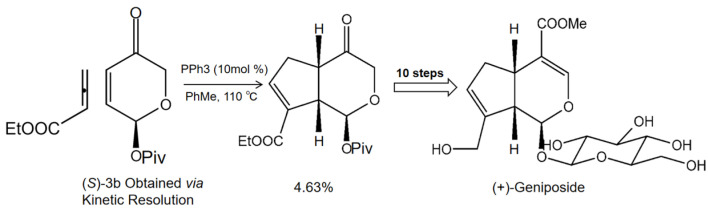
The synthetic method of natural product geniposide. Ethyl 2,3-butadienoate and enone (*S*)-3b undergo [3+2] cycloaddition reaction under phosphine catalysis to generate cis-fused cyclopenta[c]pyran 4; then, the cis-fused cyclopenta[c]pyran 4 is converted to the natural product iridoid glycoside (+)-geniposide in 10 steps [14].

**Figure 2 molecules-27-03319-f002:**
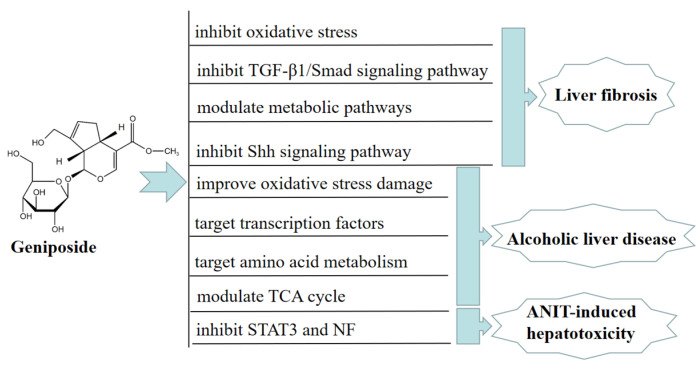
A summary of mechanisms of geniposide on liver injury disease.

**Figure 3 molecules-27-03319-f003:**
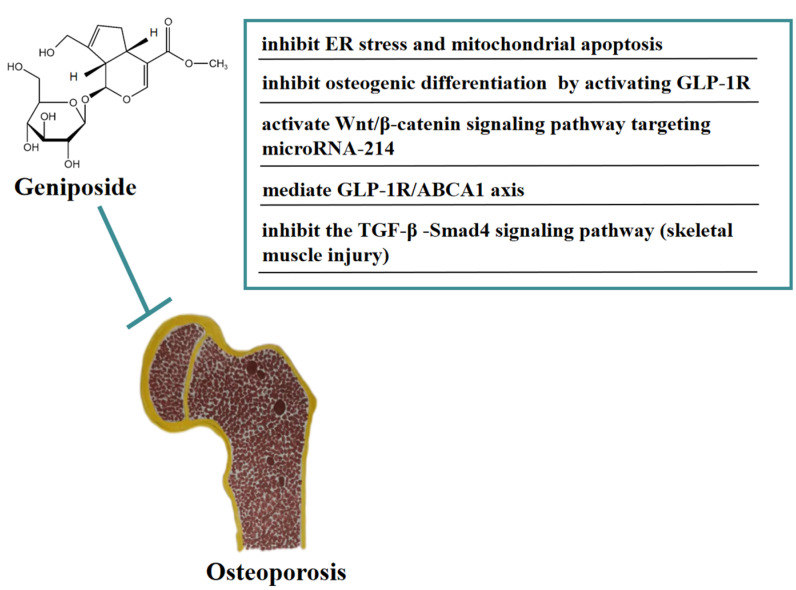
A summary of mechanisms of geniposide on osteoporosis.

**Figure 4 molecules-27-03319-f004:**
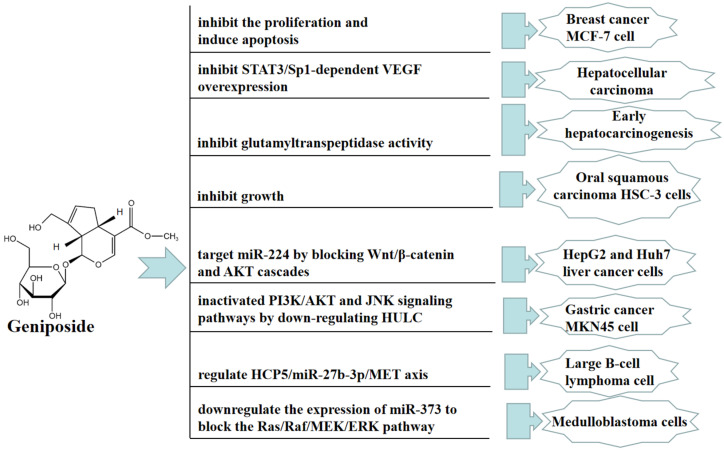
A summary of mechanisms of geniposide on tumors.

**Figure 5 molecules-27-03319-f005:**
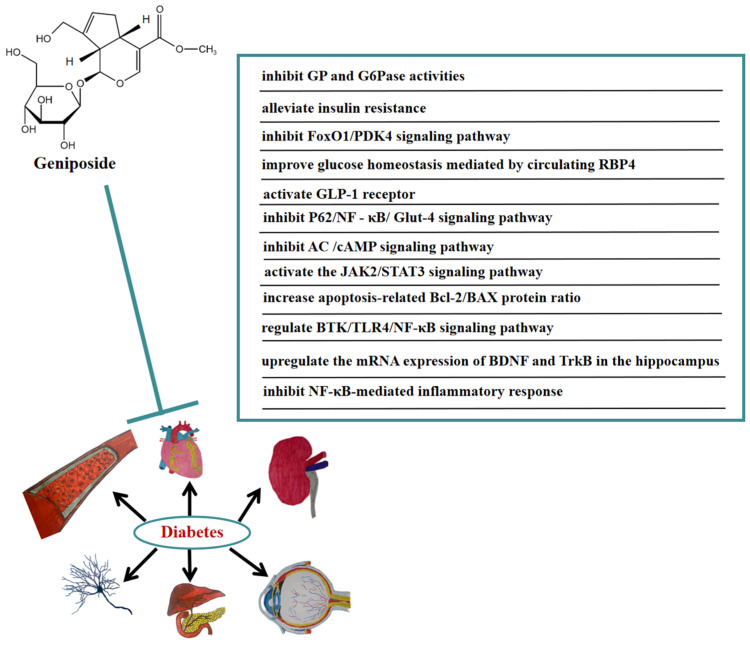
A summary of mechanisms of geniposide on diabetes.

**Figure 6 molecules-27-03319-f006:**
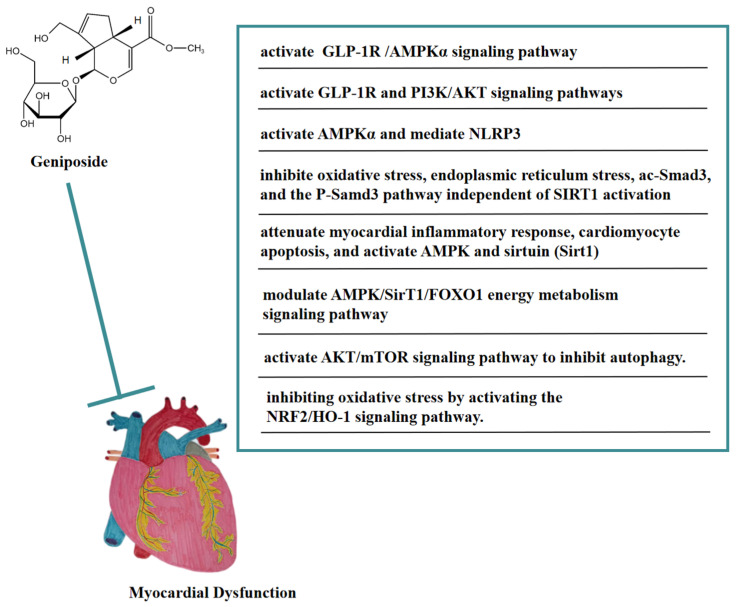
A summary of mechanisms of geniposide on myocardial dysfunction.

**Figure 7 molecules-27-03319-f007:**
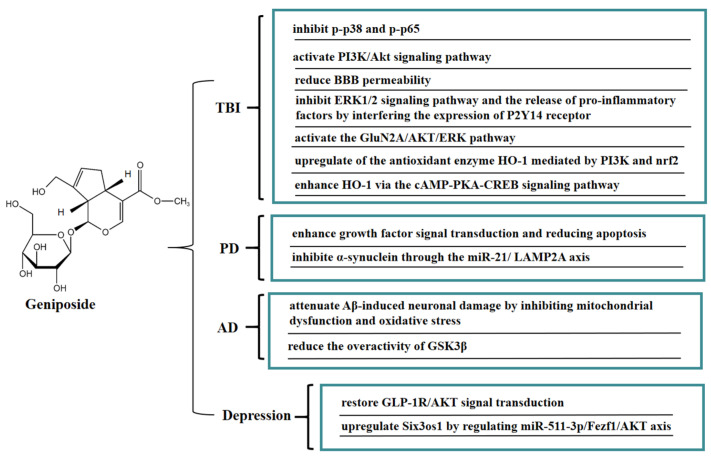
A summary of mechanisms of geniposide on neurological disease.

**Figure 8 molecules-27-03319-f008:**
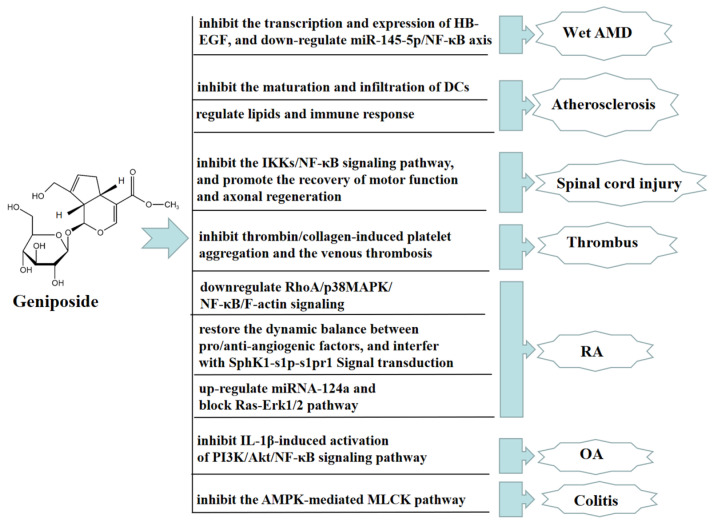
Other therapeutic effects of geniposide.

## Data Availability

Not applicable.

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
