# Peer review of "Updated Pharmacological Effects, Molecular Mechanisms, and Therapeutic Potential of Natural Product Geniposide"

_molecules, 2022, doi:10.3390/molecules27103319_

Round 1

Reviewer 1 Report

The manuscript presents a good review of the pharmacological effects and molecular mechanisms of geniposide. However, in the introduction section, a brief comment on the main methods of synthesis of the aforementioned compound is missing. For example see: Org. Lett. 2009, 11, 8, 1849–185. I accept the manuscript with a minority review that should include the main methods of synthesis of geniposide.

Author Response

The manuscript presents a good review of the pharmacological effects and molecular mechanisms of geniposide. However, in the introduction section, a brief comment on the main methods of synthesis of the aforementioned compound is missing. For example see: Org. Lett. 2009, 11, 8, 1849–185. I accept the manuscript with a minority review that should include the main methods of synthesis of geniposide.

Response:We thank the reviewer for raising this question, we have added the current main synthesis method of geniposide in the introduction section ( page2 line62-line68, Figure 1).

Reviewer 2 Report

The manuscript entitled “Updated Pharmacological Effects, Molecular Mechanisms, and 2 Therapeutic Potential of Natural Product Geniposide” provides insights into the activities and therapeutic potential of an iridoid derivative, geniposide.

Although manuscript has interesting contents, I suggest authors to revise thoroughly based on following points:

1) Manuscript has high similarity with previously published documents. Specially, few lines in abstract are completely copied (report attached). They should be revised.

2) Scientific names of plants and microorganisms should be in italics Eg .Line 20, 38, 39

3) Gardenia jasminiodes is one of the sources of geniposide. As this manuscript focuses on the specific compound, the occurrence of geniposide in different plants sources should be clearly mentioned along with its structurally similar compounds.

4) Geniposide is a glucoside of genipin. When administered in human body, it might be hydrolyzed to genipin and/or might be metabolized to other derivatives in body. Authors do not explain about it. Are these activities due to geniposide or its aglycone, genipin?

5) When authors are focusing on therapeutic potential, pharmacokinetics of geniposide should be elaborated in separate section. Is it absorbed in our body? How is the solubility? What are metabolites, etc.?

6) The safety profile and toxicity should be clearly mentioned in separate section.

7) One of the main issues is that, the authors have mostly collected information from in vitro (cell based) assay? Will it have same effect on in vivo systems and clinical studies? Thus, in vitro and in vivo studies should be separately explained as subsections for each disease category.

8) Clinical studies should be explained in separate section. Are there any clinical trials that provide sufficient evidence?

Author Response

Major comments

Although manuscript has interesting contents, I suggest authors to revise thoroughly based on following points:

1) Manuscript has high similarity with previously published documents. Specially, few lines in abstract are completely copied (report attached). They should be revised.

Response:We thank the reviewer for pointing this out. We have rewritten the abstract section according to the plagiarism report of manuscript, and other similar parts of the manuscript have been revised.

2) Scientific names of plants and microorganisms should be in italics Eg .Line 20, 38, 39

Response:Thank you for your valuable advice. We have italicized the scientific names of plants and microorganisms in the revised manuscript.

3) Gardenia jasminiodes is one of the sources of geniposide. As this manuscript focuses on the specific compound, the occurrence of geniposide in different plants sources should be clearly mentioned along with its structurally similar compounds.

Response:Thank you for your valuable advice. We have supplemented the above content in the Introduction section ( page2 line50-line62).

4) Geniposide is a glucoside of genipin. When administered in human body, it might be hydrolyzed to genipin and/or might be metabolized to other derivatives in body. Authors do not explain about it. Are these activities due to geniposide or its aglycone, genipin?

Response:We thank the reviewer for raising this question. We have added the pharmacokinetic studies of geniposide, which initially elucidates the in vivo hydrolysis of geniposide to genipin and/or metabolism to other derivatives. According to the pharmacokinetic study of geniposide, we found that most of the studies focused on geniposide, and the metabolic distribution of geniposide in the body also showed that the presence of geniposide was detected in many parts of the body (eg: A recent study found that in type 2 diabetic rats, the AUC of GJ fruit extract and pure geniposide increased similarly after oral administration, suggesting that the increase in AUC after gavage of GJ fruit extract was mainly due to the effect of geniposide, not the other ingredients in GJ. These results suggested that the pathological state of diabetes altered the pharmacokinetics of geniposide, which may be a key factor contributing to the elevation of the AUC of geniposide), especially some compound recipes containing geniposide found that the main component of Gardenia jasminoides Ellis for its efficacy is geniposide (eg: The pharmacokinetics of geniposide in compound Huanglianzhizi decoction or Huanglianjiedu decoction in middle cerebral artery occlusion (MCAO) model rat analysis showed that compared with the control group, the AUC and Cmax of the two compound treatment groups showed a significant increase trend. As a result, the MRT data of MCAO was larger, indicating that under the pathological state of MCAO, the absorption of geniposide in the compound increased, the elimination rate slowed down, and the residence time in the body was longer).

5) When authors are focusing on therapeutic potential, pharmacokinetics of geniposide should be elaborated in separate section. Is it absorbed in our body? How is the solubility? What are metabolites, etc.?

Response:Thank you for your valuable advice. 

  1. We have addedthe pharmacokinetics of geniposide in separate section ( page12 line434-line529).
  2. The Pharmacokinetic study of geniposide showed that it can be absorbed in vivo. And it is easily soluble in water (solubility H2O: ≥5mg/mL ). The urine analysis results of geniposide metabolites in rats by intragastric administration showed that the main metabolic pathway of geniposide in vivo occurred after deglycosylation, followed by glucuronidation and cleavage of the pyran ring. In vitro experiments further confirmed that the main metabolite of geniposide is the glucuronic acid conjugate of genipin [104]. β-glucosidase has good affinity and catalytic efficiency for geniposide, and can hydrolyze geniposide to genipin [105]. While genipin can be further metabolized to Genipin-1-o-glucuronic acid and genipin-monosulfate under the action of enzymes [106]. (page13 line502-509)
  3. A total of 33 metabolites of geniposide were found in SD male rats after oral administration of geniposide. The tissue distribution of geniposide metabolites showed that they were mainly distributed in rat heart (6 metabolites), liver (12 metabolites), spleen (3 metabolites), brain (6 metabolites), and lung (6 metabolites), kidney (12 metabolites), rat liver microsomes (RLMs) (4 metabolites). Moreover, the in vivo metabolism of geniposide mainly involved the hydrolytic pathway of C-1 and the pathway of parent drug or isomer including demethylation, glucosylation and so on. While metabolite profiles of geniposide differ significantly in different disease models. For instance, A recent study found that in type 2 diabetic rats, the AUC of GJ fruit extract and pure geniposide increased similarly after oral administration, suggesting that the increase in AUC after gavage of GJ fruit extract was mainly due to the effect of geniposide, not the other ingredients in GJ. These results suggested that the pathological state of diabetes altered the pharmacokinetics of geniposide, which may be a key factor contributing to the elevation of the AUC of geniposide. However, another study has analyzed the tissue distribution of geniposide and its four major metabolites (G1-G4) after oral administration of geniposide in AA rats. The analysis results showed the plasma of AA rats mainly distributed geniposide, genipin (G1), the mono-glucuronide conjugate of genipin (G2), mesenteric lymph node (MLN) mainly distributed geniposide and G2, and spleen mainly distributes geniposide, G2, a cysteine conjugate ring-opened genipin ( G3) and an oxidation of G3 (G4). While GE, G1, G2 and G4 are mainly distributed in urine, and only geniposide was distributed in liver and synovium.

6) The safety profile and toxicity should be clearly mentioned in separate section.

Response:Thank you for your valuable advice. We have added the safety profile and toxicity of geniposide in the revised manuscript ( page14 line531-line573).

7) One of the main issues is that, the authors have mostly collected information from in vitro (cell based) assay? Will it have same effect on in vivo systems and clinical studies? Thus, in vitro and in vivo studies should be separately explained as subsections for each disease category.

Response:Thank you for your valuable advice. Since geniposide has not yet a clinical drug, no clinical research report has been reported in the literature. Therefore, our manuscript mainly reported the therapeutic effect and in-depth mechanism of geniposide on animal disease models, analyzed the molecular mechanism of geniposide in vitro. The molecular mechanism of action of geniposide may be that the in vivo research was not clearly written in the previous manuscript, so we reorganized and wrote the in vivo research, and separately wrote the in vivo and in vitro studies of geniposide according to your comments. We have revised the text to address your concerns and hope that it is now clearer.

8) Clinical studies should be explained in separate section. Are there any clinical trials that provide sufficient evidence?

Response:We thank the reviewer for raising this question. Considering the reviewer's suggestion, we have further reviewed the clinical literature on geniposide, and we are very sorry that we have not found any clinical research report on geniposide. It may be because geniposide is not a drug for clinical treatment at present.

Round 2

Reviewer 2 Report

Authors have revised the manuscript and replied to all querstions raised.